# Pre-Pandemic Cross-Reactive Immunity against SARS-CoV-2 among Central and West African Populations

**DOI:** 10.3390/v14102259

**Published:** 2022-10-14

**Authors:** Marc Souris, Léon Tshilolo, Daniel Parzy, Line Lobaloba Ingoba, Francine Ntoumi, Rachel Kamgaing, Moussa Ndour, Destin Mbongi, Balthazar Phoba, Marie-Anasthasie Tshilolo, René Mbungu, Martin Samuel Sosso, Nadine Fainguem, Tandakha Ndiaye Dieye, Massamba Sylla, Pierre Morand, Jean-Paul Gonzalez

**Affiliations:** 1UMR Unité des Virus Émergents (UVE Aix-Marseille Univ-IRD 190-Inserm 1207), 13005 Marseille, France; 2Centre Hospitalier Mère-Enfant Monkole, Kinshasa, Democratic Republic of the Congo; 3Centre de Formation et d’Appui Sanitaire (CEFA), Mont-Ngafula, Kinshasa BP 817, Democratic Republic of the Congo; 4Defideva/Altadeva—Biological Lab (Inpack), Luminy Biotech, 13009 Marseille, France; 5Fondation Congolaise pour la Recherche Médicale and Faculté des Sciences et Techniques, Brazaville, Democratic Republic of the Congo; 6Institute of Tropical Medicine, University of Tübingen, 72074 Tübingen, Germany; 7Centre International de Référence Chantal Biya (CIRCB), Yaoundé, Cameroon; 8Institut de Recherche en Santé, de Surveillance Epidémiologique et de Formation (IRESSEF), Plateforme d’Immunologie, Dakar 10200, Senegal; 9CHU Aristide le Dantec, Service d’Immunologie, Dakar 10200, Senegal; 10Laboratory Vecteurs & Parasites, Department of Livestock Sciences and Techniques, University El Hadji Ibrahima NIASSE, Kaffrine Campus, Kaffrine 24600, Senegal; 11UMI SOURCE (IRD-UVSQ/Paris-Saclay), 78280 Guyancourt, France; 12Department of Microbiology & Immunology, School of Medicine, Georgetown University, Washington, DC 20057, USA

**Keywords:** SARS-CoV-2, COVID-19, CoviDiag, natural immunity, original antigenic sin, Africa

## Abstract

For more than two years after the emergence of COVID-19 (Coronavirus Disease-2019), significant regional differences in morbidity persist. These differences clearly show lower incidence rates in several regions of the African and Asian continents. The work reported here aimed to test the hypothesis of a pre-pandemic natural immunity acquired by some human populations in central and western Africa, which would, therefore, pose the hypothesis of an original antigenic sin with a virus antigenically close to the Severe Acute Respiratory Syndrome Coronavirus 2 (SARS-CoV-2). To identify such pre-existing immunity, sera samples collected before the emergence of COVID-19 were tested to detect the presence of IgG reacting antibodies against SARS-CoV-2 proteins of major significance. Sera samples from French blood donors collected before the pandemic served as a control. The results showed a statistically significant difference of antibodies prevalence between the collected samples in Africa and the control samples collected in France. Given the novelty of our results, our next step consists in highlighting neutralizing antibodies to evaluate their potential for pre-pandemic protective acquired immunity against SARS-CoV-2. In conclusion, our results suggest that, in the investigated African sub-regions, the tested populations could have been potentially and partially pre-exposed, before the COVID-19 pandemic, to the antigens of a yet non-identified Coronaviruses.

## 1. Introduction

The emergence of the Severe Acute Respiratory Syndrome Coronavirus 2 (SARS CoV-2) and, its immediate consequence, the pandemic Coronavirus disease 2019 (COVID-19) will have been the cause, on a global scale, of disruptions in many sectors of health and societies, as well as the economy, and policies of states [1,2]. Beyond these widespread impacts of the pandemic at the global level, marked variations have been observed in the above-mentioned sectors [3] and clinical exposure depending on populations and public health systems [4]. Indeed, these short- and long-term manifestations are to be considered according to the regions and populations affected since the various factors fundamentally linked to the risk of exposure of the populations to the virus and, in the long run, to the response of the states to the pandemic. Thus, these differences are observed in several areas and in the risk of exposure and the immune response.

Indeed, more than two years after the emergence of COVID-19, in terms of the incidence of the disease, significant regional differences persist, showing the lowest incidence rates in sub-Saharan Africa, Southeast Asia, and Oceania. Such a trend was observed at the onset of the epidemic and confirmed during subsequent epidemic waves [5].

In order to explain this circumstance, several hypotheses have been proposed, including: the morbidity and mortality counts likely to be underestimated in some low- and middle-income countries due to limited epidemiological surveillance or public health screening activity [6]; the population in sub-Saharan Africa being younger, with only 2.3% of the population over 65 years of age, whereas in Europe, those over 65 years of age represent 20% of the population with more than 75% of mortality associated with COVID-19 [7]; more rural living conditions potentially increasing social distancing and reducing the spread of the disease; climatic and environmental factors unfavorable to the virus persistence and spread [8]; a natural innate non-specific immunity, or, as an original antigenic sin (OAS), an immune response secondary to a previous contact with a Sarbecovirus related to the SARS-CoV-2 coronavirus sharing common antigenic profiles.

For a brief reminder, coronavirus genome encodes four major structural proteins including the spike (S), nucleocapsid (N), envelope (E) and, membrane (M) proteins. The transmembrane glycoprotein S plays an essential role in virus-cell infection. It comprises two functional subunits: the S1 subunit binding protein, and the fusion S2 subunit. The S1 subunit is divided into an N-terminal domain (NTD) and a receptor binding domain (RBD) responsible for binding the virus to the host cell Angiotensin-Converting Enzyme 2 (ACE2) receptor binding domain [9,10]. The N protein is involved in the externalization of viral particles from the infected cell [11].

Indeed, all the human viral species (e.g., SARS-CoV-1, SARS-CoV-2) or animal species (e.g., chiropterans especially) of the Sarbecovirus subgenus have the same antigenic structure and a very close genomic material, with a homology of more than 80%.

Further, the SARS-CoV-2 genome is closely related to SARS-CoV-1 (79.6% of homology) and several antibodies covering all structural proteins of SARS-CoV (S, M, N, E) have been identified and extensively studied showing some cross-reactivity with SARS-CoV-2, as well as partial cross-neutralization of S antibodies [12]. Moreover, sera from SARS-CoV convalescent or S1 CoV-specific animal antibodies (i.e., palm civet) could neutralize SARS-CoV infection by reducing S protein-mediated SARS-CoV entry [13]. Finally, SARS-CoV and MERS-CoV show that many fragments (S1-NTD, RBD, S2) of S protein are common targets for neutralizing antibody production [14]. Ultimately, The BANAL-52 virus, isolated from chiropteran (*Rhinolophus malayanus*), presents 96.8% homology with SARS-CoV-2.

Importantly, for our antibody-antigen research of Sarbecovirus, the N protein appears more conserved across species than the S protein, while the RBD appears more conserved within the S1 unit. From the point of view of protection (i.e., neutralizing antibodies), there is a strong correlation between the levels of RBD antibodies and neutralizing antibodies to SARS-CoV-2 in humans [15].

Ultimately, unlike the highly pathogenic coronaviruses (SARS-CoV, MERS-CoV, and SARS-CoV-2), the four common coronaviruses (HCoV-229E, HCoV-OC43, HCoV-NL63, HCoV-HKU1) cause mild upper respiratory tract disease in adults that passes unnoticed in public health. Moreover, it appears that infection with these seasonal human coronaviruses does not protect against SARS-CoV-2 infection [16].

The objective of our study was to consider the major hypothesis of a pre-existing (before the COVID-19 emergence) natural humoral anti-SARS-CoV-2 immunity among African and Asian populations, to understand the origin of this potential pre-pandemic immunity, while underpinning the central idea of a potential effect on exposure/immune response to the pathogen.

To detect SARS-CoV-2 reacting antibodies, we tested pre-pandemic sera samples against five SARS-CoV-2 proteins playing an essential role in virus attachment (i.e., fusion, entry, and transmission). We present here our initial results obtained by testing 1655 sera samples collected months before the COVID-19 epidemic started from people residing in the Democratic Republic of Congo (DRC), Cameroon, Republic of Congo (ROC), and Senegal.

## 2. Materials and Methods

### 2.1. Antibody Detection and Assay Specificity

We used an innovative multiplex immunoassay from INNOBIOCHIPS company (CoViDiag^®^ assay). This qualitative miniaturized and parallel-arranged enzyme-linked immunosorbent assay (ELISA) detected simultaneously the presence of IgG antibodies directed against five SARS-CoV-2 antigens in human serum or plasma: the N protein, the S1 protein, the RBD domain of the S1 protein, the NTD domain of the S1 protein, and the S2 protein from SARS-CoV-2 virus Wuhan strain [17]. The ELISA values were obtained by optical density reading using a laser reader.

For this kind of study, the specificity of the ELISA is critical. A study of 16 serological ELISA SARS-CoV-2 assays was conducted using pre-pandemic sera from Switzerland to assess possible cross-reactivity with other human viral infections, including HCovs, Parvovirus B19, Cytomegalovirus, Epstein- Barr virus, Tick-borne encephalitis virus, Herpes simplex virus 1 and 2, Influenza A and B, Respiratory Syncytial Virus, Measles virus, Mumps virus, Rubella virus, and Varicella-zoster virus. No cross-reactions were observed [18].

In concordance with these results, the specificity of the CoViDiag^®^ assay was evaluated to 99% by INNOBIOCHIPS company, and the assay was compliant with French HAS (Haute Autorité de Santé) and regulatory demands. In particular, the specificity of the test against the known low pathogenic human seasonal coronaviruses (HCoVs: 229E, OC43, NL63, HKU1) was evaluated by INNOBIOCHIPS company using 25 pre-pandemic samples positive for HCovS [19]. All these HCOVs positive samples tested negative for all SARS-CoV-2 proteins used in the ELISA. Test specificity was confirmed by [20] using 132 non-SARS-CoV-2 sera collected in Belgium before the COVID-19 pandemic. These potential cross-reactive samples included positive antibodies against HCoVs (32 samples positive against 229E coronavirus, 118 samples positive against OC43 coronavirus, 19 samples positive against NL63 coronavirus). The authors found an excellent specificity of the CoViDiag^®^ assay for all tested antibodies (99.2 to 100%).

### 2.2. Control Samples and Assay Sensitivity Analysis

Sensitivity was analyzed using controls obtained by INNOBIOCHIPS company from 189 samples from blood donors collected in Northern France before the COVID-19 pandemic, randomly selected (EFS, Établissement Français du Sang) and tested negative for SARS-CoV-2 by RT-PCR. This sera collection was used by INNOBIOCHIPS manufacturer to define the thresholds of positivity (cutoffs) as compared to the sera collected from patients infected by SARS-CoV-2 (RT-PCR test positive).

We also used the control group from INNOBIOCHIPS manufacturer as a negative control group to evaluate the results from African countries with respect to their geographic origin (Africa vs Europe). Such blood donors from France, as for most of the population in Europe, are supposed to have not been in an environment potentially favorable to the direct or indirect contact with bats. As INNOBIOCHIPS company, we used this control group to establish thresholds (cutoffs) for the absence of SARS-CoV-2 antibodies in the ELISA. To eliminate the risk of false negatives (samples may came from donors of African or South and South-East Asian origin), for each antigen the distribution of control values was modeled (negative exponential distribution for N, S2, RBD, NTD, Weibull distribution for S1). For each antigen, the threshold value we used to characterize pre-pandemic positive samples (namely PRECOV threshold) correspond to a probability equal to 0.0002 for a control sample to be a false negative: a control sample with a value for an antigen above the threshold will, therefore, be considered as a false negative for this antigen.

CoViDiag^®^ assay sensitivity was also evaluated and optimized by [20] using 135 sera from 94 SARS-CoV-2 RT-PCR positive patients from Belgium. The results of this study were consistent with the PRECOV thresholds defined in our study.

### 2.3. Data Analysis

After calculating the statistical moments and the distribution of the samples’ values for each antigen, several statistical tests and calculations were performed: 

For each antigen, the comparison of the means (Student’s *t*-test) and the variances (F-test) between the samples group and the “control” group.

Difference between the two groups (samples and controls) considering all five antigens together was tested using Hotelling test; the control group was taken as a whole, without excluding the few suspected false-negative samples. 

Calculation of confidence interval was completed for the number and percentage of samples considered positive, according to the cutoff value, for each antigen; calculation of confidence interval for the number of positive samples was made for two or more antigens.

All data are available in the main text or in the Appendix A.

### 2.4. Sample Sera Collection

We tested a total of 1655 samples from DRC (Democratic Republic of the Congo), Congo, Cameroon, and Senegal, to cover different types of environments in west and central Africa. DRC samples originated from the Monkole Hospital Center biobank (190 samples, collected in 2019 from healthy subjects from the hospital staff, from volunteers, and from young sickle-cell disease patients who were part of a study cohort), and from the ALTADEVA/Monkole biobank (384 samples, collected in 2014 and 2015 as part of a study of *Plasmodium falciparum* chemoresistance in the city-province of Kinshasa, in the central province of Kongo, and southwestern DRC).

The samples from Cameroon (383 tested) were selected from multiple collection sources, as part of continual HIV monitoring control program (Programme de Lutte contre le HIV, PLHIV), from June 2018 to June 2019, and preserved at the Chantal BIYA International Research Center for HIV Prevention and Management (CIRCB), 51% of the 383 samples selected were among samples received from some Central and General Hospitals within the country. The remaining 49% were selected among samples received from peripheral healthcare facilities.

The samples from the Republic of Congo (536 tested samples) were collected by The Fondation Congolaise pour la Recherche Médicale of the Madibou Southern district of Brazzaville, and in the Northern part of the country (Sangha) of Bomassa district of, in 2016 and 2019, respectively.

The samples from Senegal (162 tested samples) were randomly selected from serums collected in 2018 and 2019 and conserved in the biobank of the Institut de Recherche en Sciences de la Santé, Epidémiologie et formation (IRESSEF, Health Sciences Research Institute of Epidemiology and Training).

All samples were aliquoted and kept frozen as appropriate and each sample had companion data, including the date of collection, age, sex, and province of origin.

### 2.5. Ethical Approvals

All samples used in this study were collected before November 2019, from volunteer donors, and originally addressed to the local laboratories of the partner institutes for diagnostic purposes. Blood samples were collected after informed consent for the use and reuse from each patient or from his or her parent/guardian in the case of minors. All experiments were performed in accordance with relevant named guidelines and regulations. All documents and samples were anonymized. For samples from DRC, ethical approval was obtained from the Centre de Formation et d’Appui Sanitaire/Centre Hospitalier Monkole (N/Ref.: 01/CEFAMONKOLE/CEL/2013) ethics committee. For samples from Republic of Congo, ethical approval by the Fondation Congolaise pour la Recherche Médicale (Congolese Foundation for Medical Research) Institutional Review Board (IRB) and ethics committee (N/Ref.: 001/CEI/FCRM/2012 and 019/CEI/FCRM/2018). For samples from Cameroon, full approval was obtained from the IRB of the Centre International de Référence Chantal Biya (Chantal Biya International Reference Center). Full approval was obtained from the IRESSEF IRB (N/Ref.: Protocole SEN/20/30) for the Senegalese samples.

## 3. Results

The control samples from the French blood donors tested for SARS-CoV-2 reacting antibodies are shown in Table 1. Among 189 control samples, we detected 18 samples with antibodies reacting against at least 1 antigen (9.5%), and 0 with antibodies reacting against at least 2 antigens (0%).

Antibodies against the five tested SARS-CoV-2 antigens were detected in the African pre-COVID samples, with differential optical density mean values significantly higher than for the control samples (Table 2). The S1 antigen showed the highest percentage of positives: 19.64% for African samples versus 2.11% for the control samples. The S2 and RBD antigens also showed significantly higher rates. We also found a high significant difference with control samples (*p*-value < 10^−6^) when all antigens were considered together.

Among the 1655 tested samples, 630 samples reacted against at least 1 antigen above the threshold (38.1%, vs. 9.5% for the controls), while 205 samples reacted against at least 2 antigens above the threshold (12.4%, vs. 0% for the controls). Serological responses of these pre-COVID African samples did not differ by age or sex.

The results observed by country (Table 3) showed higher values for Cameroon, especially for S1, S2, and S1-RBD antigens. The other antigens (N, S1-NTD) did not show significantly different values between countries. All values were significantly higher than control values.

## 4. Discussion

Our major inquiry in this discussion is to understand the origin of this observed antigenic cross-reaction of SARS-CoV-2 in pre-pandemic sera in the African countries of our study. 

Specific antibodies against SARS-CoV-2 proteins in sera collected in sub-Saharan African regions before the emergence of SARS-CoV-2 in China have already been reported by other studies, with higher serological cross-reactivity to SARS-CoV-2 in sub-Saharan African regions than elsewhere [21,22], and sometimes attributed to higher exposure to human seasonal coronaviruses (HCoVs) in these regions [23].

Although it has been shown that cross-reactive T cells against SARS-CoV-2 can be induced by common cold coronavirus, by SARS-CoV, or eventually by other animal Betacoronaviruses [24,25,26,27], as already mentioned, the ELISA we used was very specific for SARS-CoV-2 and did not detect any antibodies to seasonal HCoVs (229E, OC43, NL63, HKU1), and confirmed that the identified antibodies cross-react precisely against the SARS-CoV-2 proteins.

We observed a stronger cross-reactivity to S1, S2, and RBD (considered specific to SARS-CoV-2) than for the N protein, considered as a common response to Betacoronaviruses. Several authors indicated that antibody responses against N appeared to wane in the post-infection phase, where S protein antibody responses persisted over time, and moreover to conclude that the anti-S antibody response might be more specific, as the N protein of the SARS viruses [28,29]. Indeed, while the N protein was the most conserved, the S1 subunit was the least conserved and cross-reactivity could not be explained by exposure to the known HCoVs. Therefore, it could be suggested that this cross-reactivity must have been induced by a virus similar (i.e., sharing S epitopes) to SARS-CoV-2, rather than by any other already known human Betacoronaviruses of the Sarbecovirus sub-genus. Moreover, such cross-reactivity with S1 did not quantitatively reproduce the S1-RBD or S1-NTD data. These consistent discrepancies could be explained by a variation of antibodies affinities to these epitopes due to a change of the structure of the S protein or a change in its amino-acid sequence. Eventually a more consistent response with S1 could be the fact of a non-tested here CTD protein [30]. Altogether, such discrepancy of antibody response was in favor of an S1 belonging to a Sarbecovirus-like virus, while all sequences of the antigens included in the commercial CoViDiag^®^ ELISA were entirely based on the reference SARS-CoV-2 Wuhan strain. It was also important to consider that S and N protein sequences were equally divergent among coronaviruses, while the S2 subunit was better conserved than the N protein. Ultimately the RBDs of these viruses slightly differed from that of SARS-CoV-2 and bound as efficiently to the hACE2 protein as the SARS-CoV-2 Wuhan strain isolated in early human cases. Moreover, it was established that the RBD as a human-ACE2-dependent mediator in human cells was inhibited by antibodies neutralizing SARS-CoV-2 [31,32].

Considering this, even though anti-HCoV antibodies were known to cross-react with SARS-CoV-2 (humoral and tissue), it was clearly demonstrated in large cohorts that antibodies against seasonal HCoVs did not produce any protection against SARS-CoV-2 [33,34,35].

Therefore, our results were in favor of the hypothesis that some populations in Africa and potentially from other part of the World (e.g., South East Asia) might be less susceptible to SARS-CoV-2 infection due to a pre-existing immunity triggered by other not yet identified Betacoronavirus, potentially of an animal origin, probably chiropteran. Such type of antigenic relationship and acquisition of natural immunity without morbidity, have already been observed with several viruses from various virus families, including filoviruses and flaviviruses [36,37].

Sarbecovirus sub genus of the Betacoronavirus genus includes numerous virus species of several Chiropteran species. Chiropteran insectivorous species, as well as fruit bats, are hosts for several coronaviruses’ species close to the original strain of SARS-CoV-2 isolated in China [38]. SARS-CoV and MERS-CoV viruses are also monophyletically placed with chiropteran coronavirus parental species [39,40]. In Cambodia and Myanmar, viruses closely related to SARS-CoV-2 were isolated from bat samples (*Rhinolophus shameli*) collected before 2020 [41,42,43]. Many species of Rhinolophus and other Microchiroptera species that were hosts of Betacoronavirus and potentially reservoir of SARS-CoV related viruses could be found in Europe and Africa, and even on the Australian mainland [44,45,46], but globally it could be observed that the spatial distribution of fruit bats [47,48] matched the spatial distribution of countries with lower symptomatic circulation of COVID-19. Many species of fruit-eating chiropterans have been known to occur in Africa and have been found to carry a significant variety of Betacoronaviruses related to SARS-CoV-2. The ecology and behavior of bats, especially fruit bats (e.g.: mass frequentation of fruit orchards, roosting trees close to dwelling) may have favored direct or indirect contact with humans, and such mostly in rural population of Africa and Asia, where the probability of contact with chiropterans was higher. Several Betacoronaviruses similar to the Sarbecovirus cluster have been found in horseshoe bats, as well as specific antibody response to these viruses in African fruit bats [49,50].

Moreover, a new bat Sarbecovirus (namely Bana virus) isolated from Microchiroptera (i.e., insectivorous bats) in Laos seemed to have the same potential for infecting humans as early strains of SARS-CoV-2 and for being, so far, the closest strain phylogenetically [31]. Furthermore, the authors showed that the S protein of SARS-CoV-2 was a mosaic of sequences extremely similar to the one of this new bat virus isolated. This allowed for consideration of the analogy that the African populations could also be exposed to an African Microchiroptera Sarbecovirus, which was yet to be discovered or identified, knowing that these bats were also very common in this continent [48,49,50,51]. Moreover, coronaviruses were detected from several bat species in close contact with human in Rwanda (Africa) and Madagascar, including known and novel Betacoronaviruses genetically close to the SARS-CoV-2 [49,50,51,52].

We are aware of the limitations of our study and then, given the novelty of our results, it is necessary to confirm it in further studies. Therefore, as a continuation of this study, the next step is to conduct discrete surveys among indigenous populations potentially and naturally exposed to an S1 antigenic motif. To this end, back-to-back testing with the present ELISA and a SARS-CoV-2 neutralization test will be conducted to confirm the surprisingly high S1 protein cross-reactivity and, the potential of pre-pandemic neutralizing antibodies against SARS-CoV-2 infection. In addition, if such neutralizing antibodies are detected with a high value, especially in central African populations, this will again raise the question of possible cross-reactivity with other microorganism. Indeed, we are well aware that this could be due not only to widespread circulation/exposure of Betacoronavirus from chiropterans or other animal hosts, but also to potential cross-reactivity induced by other microorganisms (e.g., malaria, tuberculosis, etc.) as previously observed but not fully understood [53,54,55,56].

What is to be retained, if our observations were confirmed, is that seroprevalence studies in Africa (especially those using the S1 protein) are over-estimating SARS-CoV-2 circulation. Above all, vaccination campaigns, and eventually the development of new vaccines could consider this potential original antigenic sin (OAS) in certain populations. Thus, on the one hand, vaccine strategies, particularly in low- and middle-income countries, could target populations with no or low pre-immune response to SARS-CoV-2 and effectively reduce the risk of transmission in these vulnerable populations. On the other hand, vaccines against SARS-CoV-2 variants will be able to consider this OAS to evaluate their efficacy and, using the knowledge acquired by the identified natural antibodies, to target responsible viral proteins or antigenic sites that will become preferential in the development of personalized type vaccines.

## Figures and Tables

**Table 1 viruses-14-02259-t001:** ELISA Optical density values of the of the control samples, PRECOV threshold, and percentage of control samples above the threshold. The PRECOV thresholds were determined modeling the values distribution of the control samples.

Antigen (189 Samples)	N	S1	S2	S1-RBD	S1-NTD
Min	0	0	0	0	0
Max	17.95	1.40	13.56	1.10	5.24
Mean	0.695	0.106	1.125	0.105	0.353
1st quartile	0.07	0.04	0.08	0.02	0.03
Median	0.117	0.067	0.254	0.067	0.114
3rd quartile	0.31	0.10	1.13	0.12	0.28
Standard Deviation	2.22	0.16	2.11	0.158	0.796
PRECOV Threshold	9	0.6	10	0.6	4.5
Value above Threshold	1.6% (3)	2.1% (4)	1.5% (3)	2.6% (5)	1.6% (3)

**Table 2 viruses-14-02259-t002:** Samples values, obtained by differential optical density (1655 samples). For each antigen, the table indicates the distribution of values and the number of samples with value above the threshold (percentage and 95% confidence interval, count). The *p*-value of the Student *t*-test indicates the probability of no difference between the mean of Central Africa samples and the mean of control samples.

Results	N	S1	S2	S1-RBD	S1-NTD
Minimum	0	0	0	0	0
Maximum	59.09	50.28	95.40	45.34	70.37
Mean	2.63	0.98	4.39	0.44	1.19
Standard deviation	6.86	3.79	10.65	2.11	4.40
1st quartile	0.13	0.08	0.12	0.005	0.03
Median	0.53	0.17	0.59	0.08	0.13
3rd quartile	1.63	0.47	2.86	0.23	0.44
Total ^$^	6.77 (5.6–8.0) (112) ^$^	20.18 (18.2–22.1) (334) ^$^	10.75 (9.3–12.4) (178) ^$^	11.05 (9.5–12.6) (183) ^$^	5.50 (4.4–6.6) (91) ^$^
Student *t*-test ^#^	*p* = 6 × 10^−5^	*p* = 7 × 10^−4^	*p* = 1.4 × 10^−5^	*p* = 0.0159	*p* = 0.0046

^$^: Percentage of samples above threshold, (95% C.I.), (number); #: *p*-value.

**Table 3 viruses-14-02259-t003:** Number of samples with values above the threshold, by country. For each antigen and each country, the table indicates the percentage and 95% confidence interval of samples with value above the threshold.

Country	N	S1	S2	S1-RBD	S1-NTD
Cameroon (383) *	5.7% (3.4–8.0)	32.4% (27.7–37.1)	18.0% (14.2–21.8)	22.2% (17.9–26.1)	6.0% (3.6–8.4)
Congo (ROC) (536) *	7.6% (5.4–9.8)	13.6% (10.7–16.5)	10.8% (8.2–13.4)	8.8% (6.4–11.2)	5.2% (3.3–7.1)
Congo (DRC) (574) *	7.3% (5.2–9.4)	19.2% (16.0–22.4)	7.8% (5.6–10.0)	5.7% (3.8–7.6)	5.6% (3.7–7.5)
Senegal (162) *	4.3% (1.2–7.4)	16.6% (10.9–22.3)	3.7% (0.8–66)	11.1% (6.3–15.9)	4.9% (1.6–8.2)

*: (sample size).

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
