# Peer review of "Pre-Pandemic Cross-Reactive Immunity against SARS-CoV-2 among Central and West African Populations"

_viruses, 2022, doi:10.3390/v14102259_

Round 1

Reviewer 1 Report

This manuscript claims to identify for the first time SARS CoV-2-specific humoral immune responses (IgG antibodies) in Central and Western African populations. The authors utilized an ELISA-based assay that employed 5 SARS CoV-2 antigens for the simultaneous detection of IgG responses to each protein. The authors specifically used the Covidiag assay, which has been previously evaluated and was the basis for the data generated in these studies. 

One major weakness of the current study is the reliance on a reported nearly 100% specificity of the Covidiag assay for SARS-CoV2-specific antibodies. In reviewing J Clin Med. 2020 Nov; 9(11): 3752 it is evident that the ELISA is not specific for SARS-CoV2, but rather that cutoffs were established on each antigen to permit a perceive high degree of specificity by extending false-positive sample calling by elevating positive/negative threshold values. 

Several studies have previously evaluated the cross-reactivity of African sera collected in the pre-COVID-19 pandemic era to SARS-CoV2 antigens and the other known seasonal CoV antigens. Similar cross-reactivities in non-African sera have been reported extensively in the literature (e.g. Dugas et al., 2021; Greenbaum et al., 2021; Hicks et al., 2021; Laing et al., 2020; Mateus et al., 2020; Ng et al., 2020; Ringlander et al., 2021; Sermet-Gaudelus et al., 2021; Wang. et al., 2020a). 

It is, therefore, unlikely that an ELISA designed with all 5 SARS CoV-2 antigens would be exquisitely specific to SARS-CoV2 using the Covidiag assay. The present study lacks the verification of this fact by not testing the samples against antigens from seasonal CoVs. Instead, the study relies solely on the assumption that the Covidiag assay is nearly 100% specific. This assumption leads to statistical conclusions that are faulty and not in-line with previous work that has evaluated such cross-reactivities (e.g. Viruses Nov 21;13(11):2325; disclaimer - reviewer is a co-author in this publication; Also Scientific Reports (2022) 12:12962). 

Although there is a clear interest in identifying the reasons for the relatively low burden of COVID-19 in African populations, and the growing number of reports suggesting the presence of a circulating sarbecovirus in the sub-continent, the agent, if present, remains elusive. It is, therefore, important to use caution in attributing the cause of observed cross-reactivities to an unknown SARS CoV-2-like CoV. 

This study requires significant improvement before it is published. It is critical to understand the existence of cross-reactivity in pre-COVID-19 pandemic samples with seasonal CoVs and possibly as of yet unidentified CoVs. Due to the poor understanding of these correlates cross reactivity studies with antigens from all known CoVs remains a crucial element of such studies.

Author Response

We thank the reviewer for his comments and remarks.

As requested, we rewrite the introduction, adding references and merging introduction and background.

Reviewer #1: One major weakness of the current study is the reliance on a reported nearly 100% specificity of the Covidiag assay for SARS-CoV2-specific antibodies.

Authors Answer: For this kind of study, the specificity of the ELISA is critical. The specificity of the CoViDiag® assay was evaluated to 99% by INNOBIOCHIPS company, and the assay is compliant with French HAS (Haute Autorité de Santé) and regulatory demands. They follow the regulations required by the health authorities to obtain a marketing authorization for pharmaceutical products on the one hand, and on the other hand, the specificity and safety required by the law for products intended for human clinical diagnosis (which is well documented - sensitivity - specificity - on the documents provided with the product).

We have no reason to question these results, especially since they were confirmed by an independent study published in 2021. It is normal for an ELISA to use cutoffs to determine the threshold of positivity, and the values used by INNOBIOCHIPS are in fact very conservative. In addition, a study evaluating the specificity of 16 Sars-CoV-2 ELISAs on the market shows excellent specificity of the ELISA SARS-CoV-2 in general, with no test showing a reaction to HCoVs. These results are consistent with the evidence provided by the test developer. It would be interesting to test our samples for HCoVs, but as our objective is to compare pre-pandemic samples from Africa with pre-pandemic samples collected in France with respect to SARS-CoV-2 specific antigens, the fact that African samples are potentially more positive to HCoVs than European samples has no influence on our results, as the test is sufficiently specific and does not cross-react with the HCoVs eventual positivity.

We add in the article more explanation on specificity, we add references, especially in the Materials and methods section.

Reviewer #1: Although there is a clear interest in identifying the reasons for the relatively low burden of COVID-19 in African populations, and the growing number of reports suggesting the presence of a circulating sarbecovirus in the sub-continent, the agent, if present, remains elusive. It is, therefore, important to use caution in attributing the cause of observed cross-reactivities to an unknown SARS CoV-2-like CoV.

Authors Answer: We agree with this remark, and we change reference to SARS-CoV-2-like to Sarbecovirus-like.

Reviewer 2 Report

The central idea of this paper is about the potential pre-existing immunity against COVID-19 in some areas in Africa which is based on previous antigens of a COVID-like virus. This paper, for sure, have high significance, since COVID19 had caused, and is still causing problems worldwide. The overall flow of the manuscript is fine and logical. However, there are still some points that could make the paper improve. Please check the comments below:

1.     Line 47-48, better not to use the word realistic since it is away too bias.

2.     Should the background section be part of the introduction? The overall flow of the introduction section, for example, could be like: COVID causes global trouble; there is a phenomenon in specific areas; and then a major hypothesis (central idea) aiming to the phenomenon (don’t have to list everything, otherwise it’s kind of like stray from the central points); and some background information based on this hypothesis, such as what COVID protein may be the antigens (some could be taken from your background section); any other coronaviruses that is known related to COVID, and what are the similarities and differences between the related viruses (for example, you can talk from clinical symptoms and data in numbers, or from the perspective of virion structures); then talk about what’s the main hypothesis this study is trying to discuss and demonstrate (that’s the last paragraph of your current introduction section). 

3.     The whole section of your current introduction + background needs references. Please make sure that whenever you make a solid statement, put a citation afterwards.

4.     Please re-emphasize the reason why you are choosing the samples from the selected areas in Africa in results/discussion section, and why French samples could be used as controls, like what do the French samples represent, and what are the differences between the two areas according to this study could make French samples as controls?

5.     In discussion, maybe mention how could public health groups, or research groups, learn from your work? Any potential benefits on treatments or vaccines? Something like that may let readers know more about the merit of the work.

Author Response

We thank the reviewer for his comments and remarks.

Rewiever#2:  Line 47-48, better not to use the word realistic since it is away too bias.

Authors Answer: Done.

Rewiever#2:  Should the background section be part of the introduction? The overall flow of the introduction section, for example, could be like: COVID causes global trouble; there is a phenomenon in specific areas; and then a major hypothesis (central idea) aiming to the phenomenon (don’t have to list everything, otherwise it’s kind of like stray from the central points); and some background information based on this hypothesis, such as what COVID protein may be the antigens (some could be taken from your background section); any other coronaviruses that is known related to COVID, and what are the similarities and differences between the related viruses (for example, you can talk from clinical symptoms and data in numbers, or from the perspective of virion structures); then talk about what’s the main hypothesis this study is trying to discuss and demonstrate (that’s the last paragraph of your current introduction section). The whole section of your current introduction + background needs references.

Please make sure that whenever you make a solid statement, put a citation afterwards.

Authors Answer: We rewrite the introduction and background paragraph according to the reviewer comments. We add references accordingly.

Rewiever#2:  Please re-emphasize the reason why you are choosing the samples from the selected areas in Africa in results/discussion section, and why French samples could be used as controls, like what do the French samples represent, and what are the differences between the two areas according to this study could make French samples as controls?

Authors Answer: We choose the samples for selected areas in Africa to cover different types of environments in sub-sahara Africa (sub-sahel, occidental, central) where we have partners. We try now to extend this study to other countries (Ouganda, Chad, Thailand).

We add more details in the section Control samples and assay sensitivity analysis in the Materials and Method section.

Reviewer#2 :    In discussion, maybe mention how could public health groups, or research groups, learn from your work? Any potential benefits on treatments or vaccines? Something like that may let readers know more about the merit of the work.

     Authors Answer: We add in conclusion a paragraph on this matter.

Reviewer 3 Report

In this article, the authors described the Sars CoV2 IgG detected in patients before the pandemic storm in 2019. They proposed a model in which several IgG were found against S1 protein and its domains. A low cross-reactivity with a Betacoronavirus was estimated.

In my, opinion the paper is suitable for Viruses publication. Revisions are not required.

Author Response

We thank the reviewer for his positive evaluation of our work.

Round 2

Reviewer 1 Report

none

Author Response

Thank you for your review.